# Fluid Inclusion Study of the Penjom, Tersang, and Selinsing Orogenic Gold Deposits, Peninsular Malaysia

**Charles Makoundi [1],\*, Khin Zaw [1] and Zakaria Endut [2]**

[1]  Centre for Ore Deposit and Earth Sciences (CODES), University of Tasmania, Hobart 7001, Australia; KhinZaw.nilar@gmail.com

[2]  School of Materials and Mineral Resources Engineering, Universiti Sains Malaysia, Nibong Tebal, Penang 14300, Malaysia; zakaria.endut@usm.my

\*  Correspondence: c.makoundi@utas.edu.au; Tel.: +61-362262472; Fax: +61-362262547

**Abstract:** Ore-forming fluids in the auriferous district of the Central gold belt in Peninsular Malaysia were studied for their temperature, salinity, and relationship to the surrounding geology. Microthermometric analysis carried out showed homogenisation temperatures range from 210 to 348 °C (Tersang), between 194 and 348 °C (Selinsing), and from 221 to 346 °C (Penjom). Salinities range from 2.41 to 8.95 wt % NaCl equiv (Tersang), between 1.23 and 9.98 wt % NaCl equiv (Selinsing), and from 4.34 to 9.34 wt % NaCl equiv (Penjom). Laser Raman studies indicated that at the Tersang gold deposit, most inclusions are either pure or nearly pure $CO_2$-rich (87–100 mol %), except for one inclusion, which contains $CH_4$ gas (13 mol %). In addition, at Selinsing, most inclusions are $CO_2$-rich (100 mol %). However, an inclusion was found containing $CO_2$ (90 mol %), with minor $N_2$ and $CH_4$. Additionally, at the Penjom gold deposit, most fluid inclusions are $CO_2$-rich (91–100 mol %), whereas one fluid inclusion is $N_2$-rich (100 mol %) and another one has minor $N_2$ and $CH_4$. At a basin scale, homogenisation temperatures against salinity suggests an isothermal mixing of fluids. Most fluids are $CO_2$-rich and are interpreted to be of metamorphic origin. The evidence further indicates involvement of magmatic fluids that is supported by the association of sandstone and carbonaceous black shales with magmatic rocks, such as rhyolite, rhyolite-dacite, and trachyte-andesite at the Tersang and Penjom orogenic gold deposits.

**Keywords:** Peninsular Malaysia; fluid inclusions; laser Raman; microthermometry; gold deposits; orogenic

## 1. Introduction

Peninsular Malaysia is made up of three tectonic belts, the Western, Central, and Eastern belts including the northwestern (NW) domain, which comprises the Langkawi Islands (Figure 1). Most sedimentary formations have been intruded by magnetite, and ilmenite-rich granitoids. The Bentong–Raub Suture Zone transects through the Peninsula and strikes along the western margin of the central belt. Historically, the Central belt is the main gold-producing province in Peninsular Malaysia, and it is host to orogenic gold deposits [1–4].

The formation of the Bentong–Raub Suture Zone was probably coeval with the emplacement of major faults that were conduits for most of mineralising fluids. A detailed district scale fluid inclusion study has not been carried out to determine composition, temperature, and salinity information for the ore-forming fluids. A few orogenic gold deposits or intrusion-related gold deposits are found in the Central belt including Bukit Koman, Tersang, Penjom, Selinsing, and Buffalo gold deposits. Quartz veins have been sampled from high-grade zones in open pits from three selected ore deposits.

Three deposits were studied because the Bukit Koman and Buffalo Reef deposits did not provide representative drill core samples useful for fluid inclusion study. They have been analysed by microthermometric and Laser Raman techniques at the University of Canberra and the University of Tasmania in Australia.

The purpose of this paper is to document petrography of fluid inclusions of gold-bearing quartz veins, determine temperature, salinity, and composition, as well as determine the nature of intrusive rocks, which are spatially associated with sedimentary/metasedimentary rocks. Characteristics of fluid inclusion may serve as exploration tools to target mineralised zones at the mine-scale or district-scale. Findings in this study may also be applied to other auriferous districts in the world in the search for mesothermal/orogenic vein gold systems or intrusion-related gold deposits.

## 2. Geologic and Tectonic Setting

The Central belt of Peninsular is characterised by a tectonic suture called the Bentong–Raub Suture Zone, which transects shallow, and deep-marine sedimentary and volcanic rocks on the Eastern side of the Western belt (Figure 1). The Tersang, Penjom, and Selinsing gold deposits are in the Gua Musang and Semantan formations as shown in Figure 1.

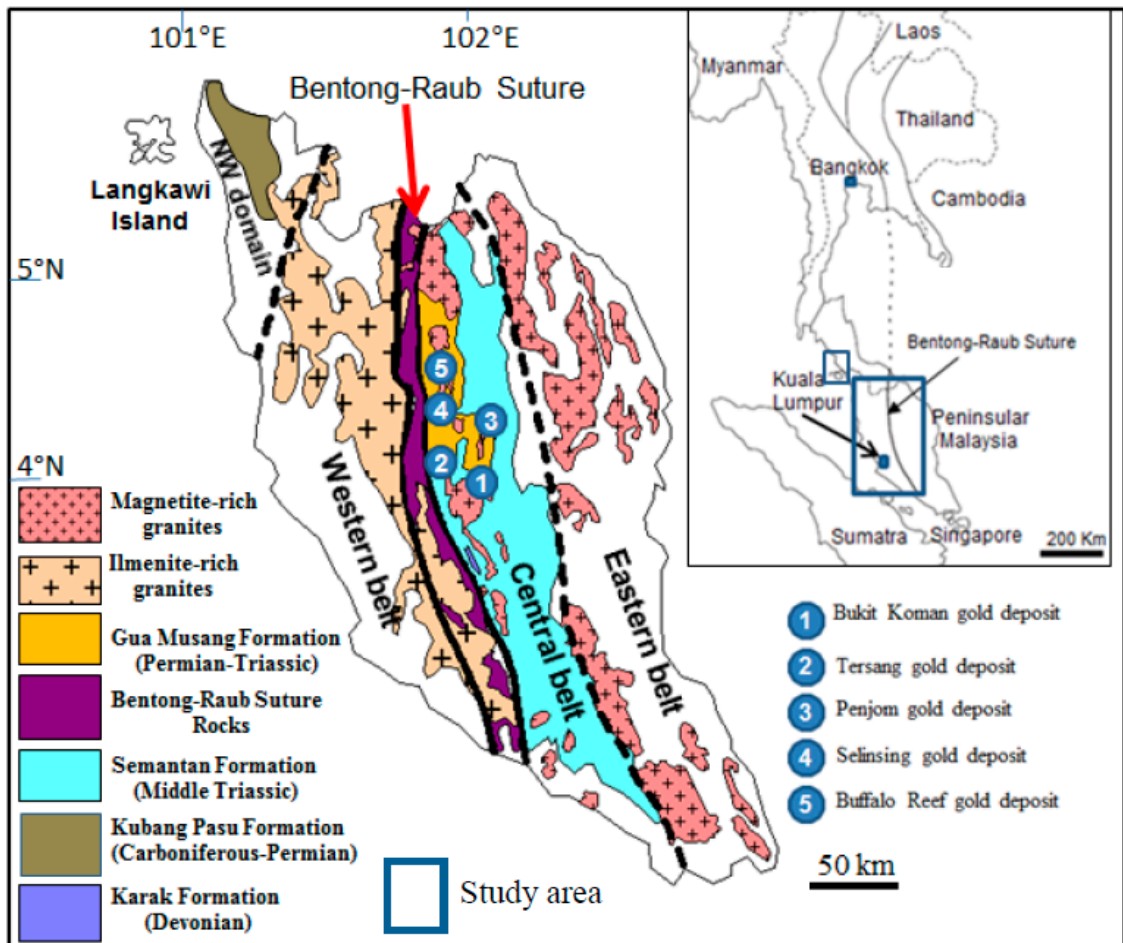

**Figure 1.** Map of Malaysia showing distribution of Phanerozoic formations, magnetite, and ilmenite-rich granites, the Bentong–Raub Suture Zone and the selected sediment-hosted gold deposits (modified after [5]), Peninsular Malaysia.

Outcrop and hand specimens of rock samples contain fluid inclusions from the Tersang, and Selinsing gold deposits, Peninsular Malaysia are presented in Figures 2 and 3.

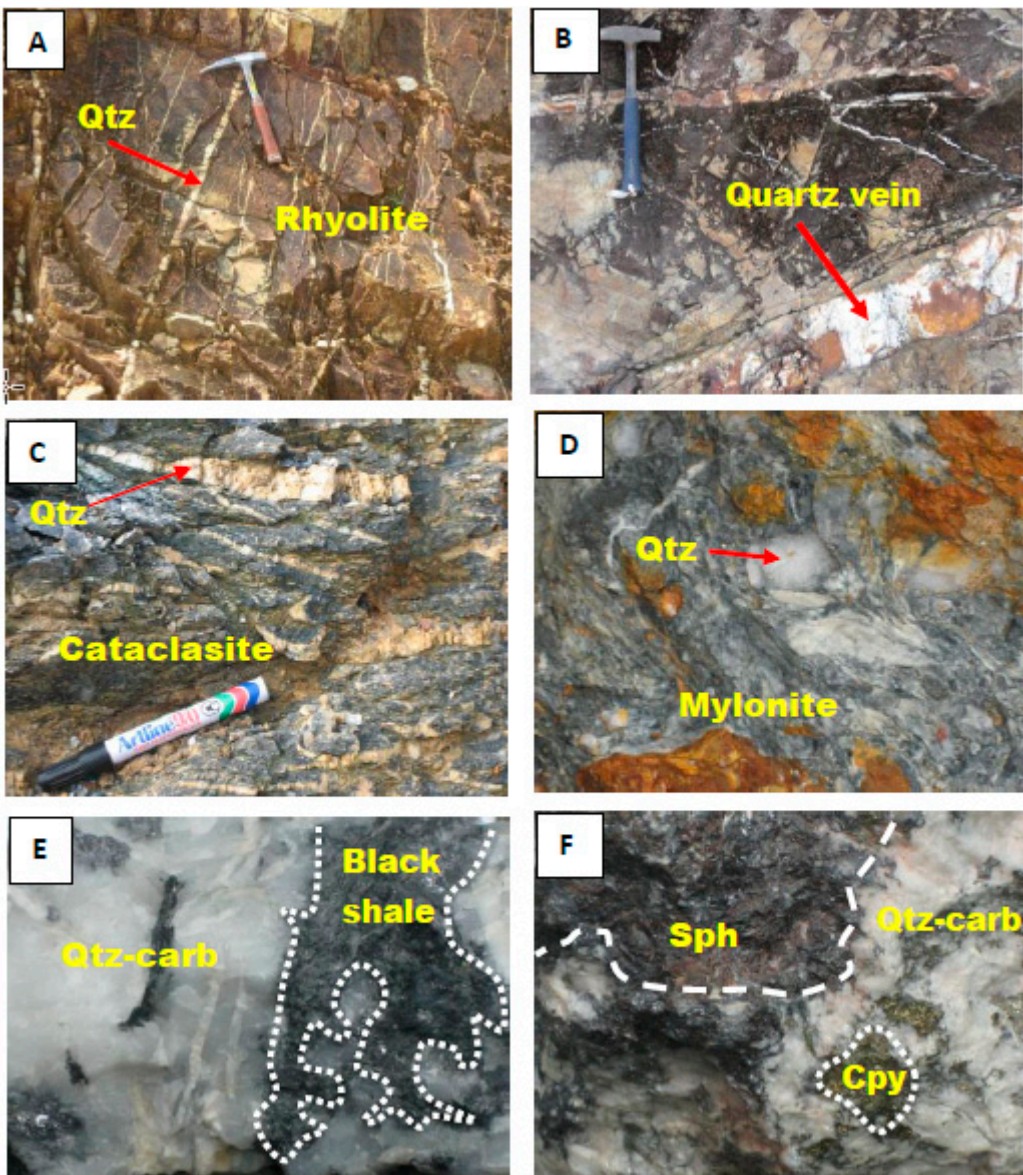

**Figure 2.** Outcrop and hand specimens of rock samples containing fluid inclusions from the Tersang and Selinsing gold deposits, Peninsular Malaysia. (**A**) Quartz sheeted veinlets at Tersang in rhyolite. (**B**) Quartz veins in sandstone at Tersang. (**C**) Deformed and lens-like quartz veins at Selinsing. (**D**) Quartz grains in mylonite at Selinsing. (**E**) Milky quartz veins, and galena in carbonaceous shales at Penjom. (**F**) Milky quartz, sphalerite, chalcopyrite in brecciated conglomerate at Penjom. Abbreviations: Qtz = quartz; Sph = sphalerite; Cpy = chalcopyrite; Carb = carbonate.

At the Tersang gold deposit, gold mineralization associated with pyrite and arsenopyrite occurs in quartz-sulphide veins hosted by grey fine-grained sandstone and alkali rhyolite sills. At the Penjom gold deposit, the gold mineralisation occurs as gold associated with carbonate-rich zones hosted within dilated quartz veins, which carry large amount of sulphides. In addition, gold is disseminated within stockwork of quartz-carbonate veins related to tonalite. Gold is often associated with arsenopyrite and pyrite in quartz-carbonate veins and stringers hosted within shear zones of brittle-ductile nature in all rock types including rhyodacitic volcanic rocks.

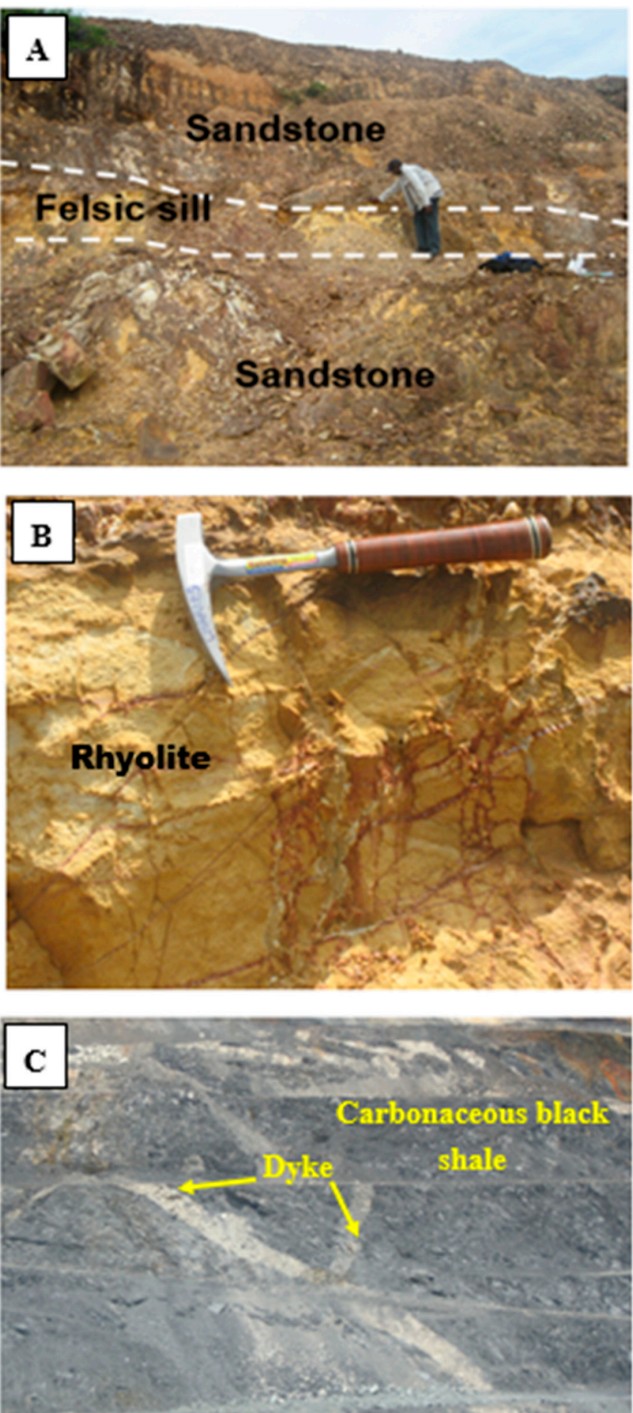

**Figure 3.** Spatial relationship between carbonaceous black shales, sandstone, and igneous rocks at the Tersang gold deposit, Peninsular Malaysia. (**A**) Felsite/rhyolite sill in contact with sandstone layers at Tersang gold deposit. (**B**) Rhyolite outcrop with quartz stringers at Tersang gold deposit. (**C**) Outcrop of carbonaceous black shales associated with igneous dykes at the Penjom gold deposit.

At the Selinsing gold deposit, gold mineralisation occurs in the form of fine gold particles commonly associated with arsenopyrite and galena hosted in cataclasite, mylonite, phyllite, sandstone, and siltstone.

### 3. Methodology

Field work was done at the Tersang, Selinsing, and Penjom open pits where gold was being mined. Quartz veins have been collected from three open pits at three different locations at mine sites. Microthermometry and Laser Raman are the two methods used in this study. The former was used to determine the homogenisation temperatures ($T_h$) and salinities and the latter was useful to determine gas composition of the fluid inclusions.

The samples for fluid inclusion analysis were prepared at Centre for Ore Deposit and Earth Sciences (CODES), University of Tasmania, Hobart, Australia. Before microthermometric analysis, the samples were examined under the microscope to document the textural characteristics of fluid inclusions, recording their different types, such as primary, secondary, and pseudosecondary.

Prior to microthermometric analysis, cathodoluminescence (CL) imaging was also undertaken on selected samples to portray growth zones in quartz veins. The growth zones are particularly useful because they contain primary fluid inclusions. In addition, CL quartz imaging also helps record the secondary fluid inclusions as they are present in healed cracks or damaged zones. At a later stage, the samples were marked in the areas of fluid inclusion assemblages and then cut into chips to a size of <1 cm in diameter, before proceeding to heating and freezing experiments.

A Linkam TH600 heating-freezing stage was utilized at 1 atm for all microthermometric analyses at CODES, University of Tasmania, Hobart, Australia. Microthermometric measurements were carried out using the analytical method detailed by [6]. Temperature measurements have an accuracy of about ±1 °C for heating and ±0.3 °C for freezing. The accuracy was insured by the triple point of $CO_2$ (−56.6 °C), the freezing point of water (0.0 °C) and the critical point of water (374.1 °C). Freezing point depressions were measured prior to heating experiments and salinity values were calculated using equation from [7]: Salinity (wt % NaCl equiv.) $= 0.00 + 1.78 \times \theta - 0.0442 \times \theta^2 + 0.000557 \times \theta^3$, where $\theta$ is the freezing point depression (FDP) temperature in °C.

Laser Raman Spectrometric analysis was undertaken on unopened fluid inclusions. It is a non-destructive method to examine components such as CO, $CO_2$, $CH_4$, $H_2$, $H_2S$, $N_2$, $NH_3$, and $SO_2$ [8,9] in fluid inclusions. The fluid inclusion samples from Tersang, Selinsing, and Penjom were analysed using a Laser Raman Spectrometry (LRS) at Geoscience Australia, Canberra, Australia. Laser Raman spectra of fluid inclusion were recorded on a Dilor® SuperLabram spectrometer comprising a holographic notch filter, 600 and 1800 g/mm gratings and a liquid $N_2$ cooled, 2000 × 450-pixel CCD detector. The inclusions were hit with 514.5 nm laser excitation from a Melles Griot 543 Series argon ion laser, using 5 mW power at the samples, and a single 30 s accumulation.

The analytical instrument is a 100x Olympus microscope objective that serves to focus the laser beam and collects the scattered light. The laser spot on the samples was about 1 μm in diameter. Wave numbers are accurate to ±1 cm$^{-1}$ as indicated by plasma and neon emission lines. Analysis of gas components, such as $CO_2$, $O_2$, $N_2$, $H_2S$, and $CH_4$ was undertaken by recording spectra from 1000 to 3800 cm$^{-1}$ using a single 20 s integration time per spectrum. Laser Raman detection limits were estimated around 0.1 mol % for $CO_2$, $O_2$, and $N_2$. For $H_2S$ and $CH_4$ the detection limit was around 0.03 mol % and errors in gas ratios was less than 1 mol %. Laser Raman spectroscopy (LRS) was used to quantitatively determine the gaseous components in fluid inclusions. The Raman spectra were typically obtained after 10 accumulations with a 5 s integration time and an estimate 5 cm$^{-1}$ spectral bandpass. The Raman spectra were calibrated using argon plasma and neon emission lines. Raman detection limits are dependent on instrumental sensitivity and the partial pressure of each gas but are estimated to be about 0.15 MPa for $CO_2$ under the conditions of this study.

To know the types of igneous rocks that intruded the sedimentary rocks at the deposits, six samples of igneous rocks were analysed by X-ray fluorescence analysis (XRF) at CODES, University of Tasmania, Hobart, Australia. In this study, there is no igneous rock data from the Selinsing gold deposit, because no exposure was found during field work. The instrument used for this analysis is PANalytical Axios Advanced X-Ray Spectrometer. It is equipped with X-Ray Tubes of 4 kW max. and a Rh anode end window. The element analysed are: F, Na, Mg, Al, Si, P, S, K, Ca, Ti, Mn, Fe, and trace elements Sc, V,

Cr, Co, Ni, Cu, Zn, Ga, Ge, As, Se, Br, Rb, Sr, Y, Zr, Nb, Mo, Ag, Cd, Sn, Sb, Te, I, Ba, La, Ce, Nd, W, Tl, Pb, Bi, Th, and U. The instrument also has the following parts: Crystals: PX-10, LiF 220, PX-1 (for F, Na and Mg), curved PE002, curved Ge111. The collimators are coarse (0.7 mm), fine (0.3 mm) with high resolution (0.15 mm). The detectors are Gas flow proportional counters with P10 gas (10% methane argon), a sealed Xe Duplex and Scintillation Counter. The sample changer is PANalytical X-Y sample changer with capacity for 96 fusion discs and 64 pills.

In sample preparation for the major elements, 32mm fusion discs was prepared at 1100 °C in 5% Au/95% Pt crucibles 0.500 g sample, 4.500 g 12–22 Flux (Lithium Tetraborate–Metaborate mix), 0.0606g LiNO$_3$ for silicates. Platinum/5% gold moulds used for cooling. Sulphide bearing samples have a different mix with more LiNO$_3$ as oxidising agent and the mix is pre-ignited at 700 °C for 10 min. Ore samples and ironstones use 12/22 flux and a higher flux/sample ratio. Dolomites and limestones need pure lithium tetraborate as a flux. Iodine vapour was used as a releasing agent to remove discs from the mould. For the trace elements, 32mm diameter pressed powder pills (10 g, 3.5 tons/cm) was used and a sample Binder polyvinylpyrrolidone—methyl cellulose (PVP-MC). Corrections for mass absorption was calculated using PANalytical Super-Q software, manufactured by PANalytical in Almelo, the Netherlands with its Classic calibration model and alpha coefficients. In house inter-element corrections were also applied. Calibration was done on pure element oxide mixed in pure silica, along with International and Tasmanian reference rocks.

## 4. Fluid Inclusion Characteristics

Fluid inclusions have been examined under a petrographic microscope at the microscopic laboratory, University of Tasmania, Hobart, Australia. At the Tersang gold deposits, fluid inclusions were found in quartz veins in the sandstone layers striking N 148°–174° and dipping 48°–70° to the northeast. Under the microscope, the characteristics of fluid inclusions from the Tersang gold deposit are shown in Figure 4. Cathodoluminescence (CL) images of quartz are shown in Figure 4A,B. The two types of fluid inclusion are the following: Type I: Primary, liquid-rich, two-phase inclusions and the size of these inclusions is between 8–10 μm (Figure 4C). Type II: Primary, vapour-rich two-phase inclusions and the size of these inclusions ranges from 5 to 10 μm and they homogenised into liquid phase. In addition, the both types of fluid inclusions are found within the growth zones in quartz veins (Figure 4D).

At the Selinsing gold deposit, fluid inclusions were found from polycrystalline quartz characterised by the presence of growth zones, which are represented by the cloudy zones rich in fluid inclusions and the clear zones poor in fluid inclusions. The polycrystalline quartz grains were found in mylonitic units which showed foliations trending N160° and dipping 64° to the east. The characteristics of the fluid inclusions from the Selinsing gold deposit in Central Malaysia are shown in Figure 5. The fluid inclusions were identified based on phases and paragenesis. The elongated primary inclusions were found in the direction of the quartz growth.

Cathodoluminescence image of quartz is shown in Figure 5A. Three types of fluid inclusions are recorded at Selinsing. Type I is primary which is liquid-rich, two-phase (L + V) inclusions, and found in mineralised quartz veins. Most inclusions show variable size ranging from 10 to 40 μm (Figure 5B). Type II is secondary, which is vapour-rich, two-phase (L + V) inclusions, and less abundant in quartz veins. The inclusion size ranges from 4 to 8 μm and homogenised into liquid phase (Figure 5C). Finally, Type III is secondary, which is liquid-rich two-phase (L + V) inclusions, mostly found along healed cracks. The size of these inclusions is less than 20 μm. These inclusions are found along sealed cracks and cross-cut early generation (Figure 5D).

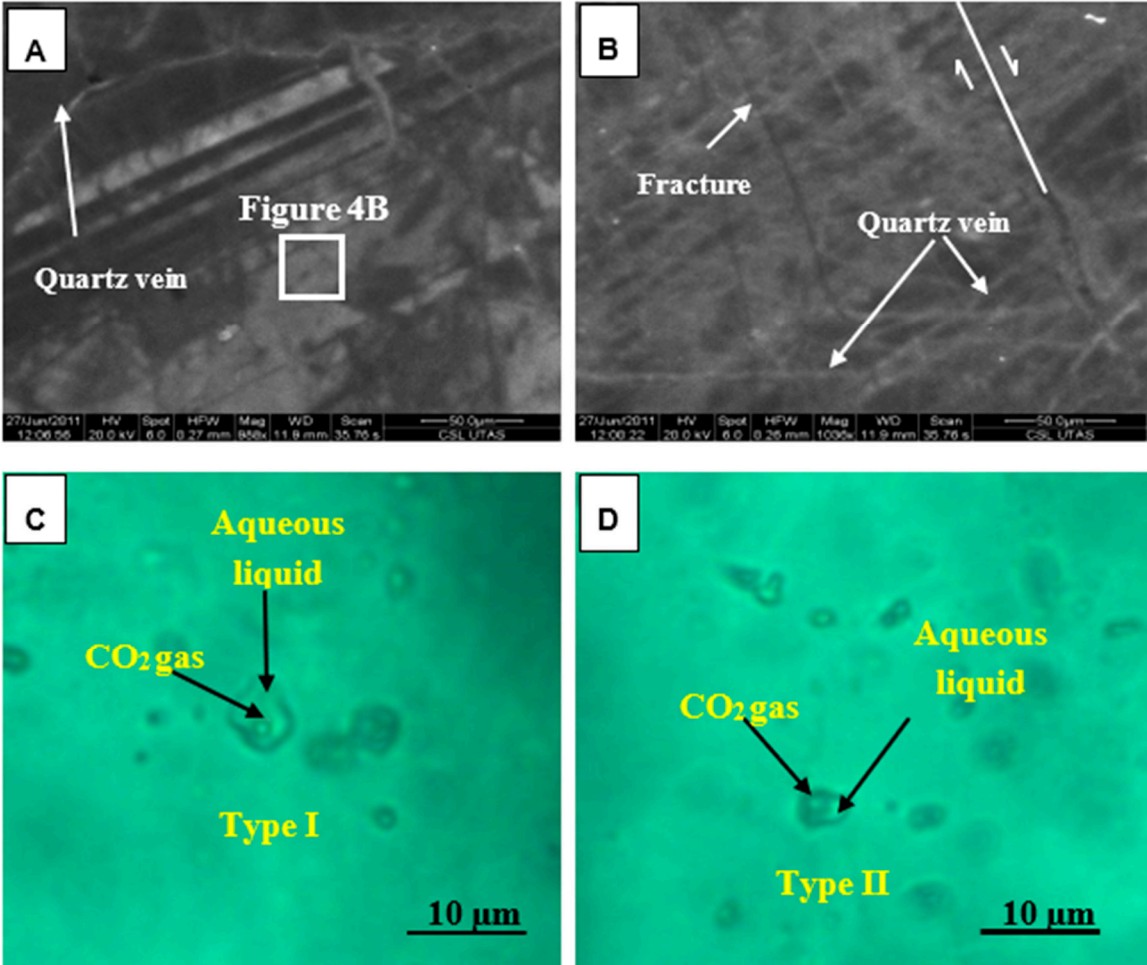

**Figure 4.** Photomicrographs of fluid inclusions in mineralised quartz veins at the Tersang gold deposit, Peninsular Malaysia. (**A**) Cathodoluminescence (CL) image showing growth zones crosscut by quartz veinlets. (**B**) CL image of growth zones disrupted by fracturation. (**C**) Liquid-rich, two-phase fluid inclusion (Type I) (Sample No. TER-R029). (**D**) Vapour-rich, two-phase inclusion (Type II) (Sample No. TER-R029).

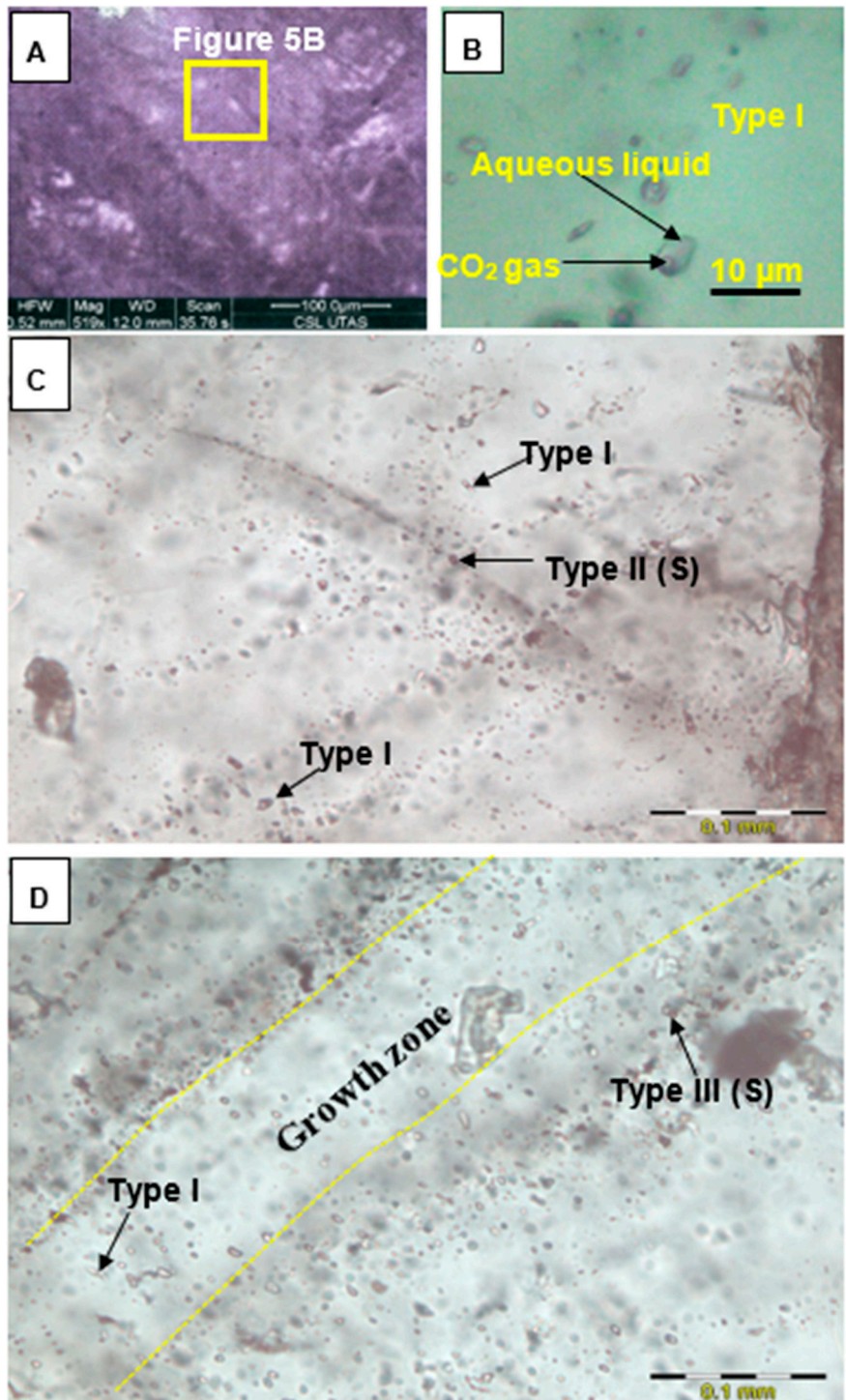

**Figure 5.** Photomicrographs of fluid inclusions in mineralised quartz veins at the Selinsing gold deposit, Peninsular Malaysia. (**A**) Cathodoluminescence image of quartz (Sample No. SEL-DD7@173m). (**B**) Primary, two-phase inclusion with liquid and gas phase (Sample No. SEL-DD7@173m). (**C**) Primary (P) and secondary (S) inclusions with liquid and vapour phase (Sample No. SEL-R032). (**D**) Secondary, two-phase (liquid + vapour) and vapour inclusions (Sample No. SEL-R032).

At the Penjom gold deposit, fluid inclusions were collected from the NS–NW trending shear quartz veins. Photomicrographs of fluid inclusions in these veins are shown in Figure 6.

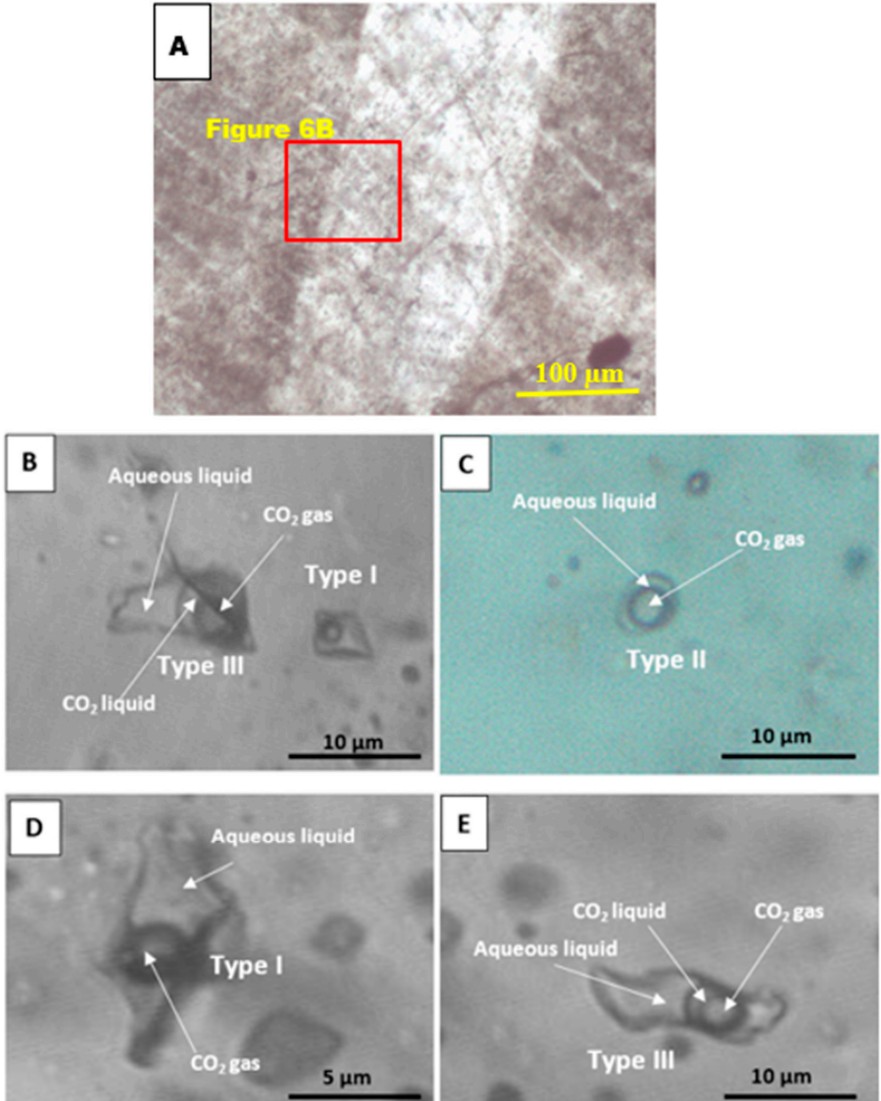

**Figure 6.** Photomicrographs of fluid inclusions in mineralised quartz at the Penjom gold deposit, Peninsular Malaysia. (**A**) Cathodoluminescence image of quartz showing growth zones (Sample No. Penjom 4). (**B**) Two, and three-phase fluid inclusions (Type I and Type III). (**C**) Vapour-rich, two-phase inclusion (Type II). (**D**) Two-phase fluid inclusion (Type I). (**E**). Three-phase fluid inclusion (Type III).

Cathodoluminescence image of quartz is shown in Figure 6A. Three types of fluid inclusions were found at the Penjom gold deposit. Type I is primary, with two-phase inclusions, liquid-rich, abundant with size ranging from 8 to 20 μm (Figure 6B). In addition, Type II is primary, two-phase, vapour-rich inclusions occurs together with liquid-rich inclusions. The size varies from 5 to 8 μm (Figure 6C,D). Furthermore, Type III is also primary, three-phase, $CO_2$-liquid bearing inclusions, and less abundant with size ranging from 10 to 15 μm (Figure 6A–E).

## 5. Results of Microthermometry

Microthermometric measurements were undertaken for the Selinsing, Tersang, and Penjom gold deposits and a summary of homogenisation temperatures and salinities data are presented in Tables 1 and 2. At the Tersang orogenic gold deposit, heating experiments indicated that the homogenisation temperatures are between 210 and 348 °C for liquid-rich (Type I) inclusions (Figure 4). Vapour-rich two-phase (Type II) inclusions are rare and no data were collected due to their smaller size for heating and freezing experiments. The salinities range from 2.41 to 8.95 wt % NaCl equiv. Other types such as

Type II, and III are excluded in this study because of their smaller size and difficulty in carrying out the heating and cooling experiments. At the Selinsing orogenic gold deposit, microthermometry was performed only for the Type I inclusions, which are liquid-rich inclusions and larger enough in size to do heating and freezing measurements (Figure 5B). Type I inclusions yielded the homogenisation temperature range from 194 to 348 °C and salinities between 1.23 and 9.98 wt % NaCl equiv. At Penjom, heating and freezing results indicated that homogenisation temperatures are between 221 and 346 °C and the salinities range from 4.34 to 9.34 wt % NaCl equiv. For the Penjom orogenic gold deposit, heating experiments were conducted for Type I, Type II, and Type III inclusions (Figure 6), but the freezing measurements were only undertaken for the Type III because other inclusion types (types I and II) were too small, and just a few to determine salinity.

**Table 1.** Summary of microthermometric data at the Tersang, and Selinsing Au deposits, Peninsular Malaysia.

| Location | Tf (°C) | FPD (°C) | Th (°C) | Salinity * wt % NaCl Equiv. |
|---|---|---|---|---|
| | nd | nd | 237.5 | nd |
| | nd | nd | 269.6 | nd |
| | nd | nd | 246.4 | nd |
| | nd | nd | 263.2 | nd |
| | nd | nd | 276.1 | nd |
| | nd | nd | 232.2 | nd |
| | nd | nd | 247.5 | nd |
| | nd | nd | 250.1 | nd |
| | nd | nd | 290.7 | nd |
| | nd | nd | 261 | nd |
| | nd | nd | 237.5 | nd |
| **Tersang Au deposit** | nd | nd | 269.6 | nd |
| **(Fluid inclusion Type I)** | nd | nd | 246.4 | nd |
| | nd | nd | 263.2 | nd |
| | nd | nd | 276.1 | nd |
| | −32.8 | −2.1 | 232.2 | 3.55 |
| | −35.3 | −1.5 | 247.5 | 2.57 |
| | −38.8 | −6 | 250.1 | 9.21 |
| | −36 | −1.4 | 290.7 | 2.41 |
| | −36.3 | −5.8 | 261 | 8.95 |
| | nd | nd | 284.2 | nd |
| | nd | nd | 270.4 | nd |
| | nd | nd | 294.6 | nd |
| | nd | nd | 348 | nd |
| | nd | nd | 283.6 | nd |
| | −37 | −5 | 203.5 | 7.86 |
| | −33 | −0.7 | 193.9 | 1.23 |
| | −37.7 | −6.5 | 213 | 9.86 |
| | −34.1 | −1.4 | 347.5 | 2.41 |
| **Selinsing Au deposit** | nd | nd | 301 | nd |
| **(Fluid inclusion Type I)** | −31.7 | −6.6 | 337 | 9.98 |
| | −40.1 | nd | 343.7 | nd |
| | −36.7 | −3.4 | 200.5 | 5.56 |
| | −37.3 | −5.7 | 216.6 | 8.81 |
| | −37.3 | −6 | 244.9 | 9.21 |

Note: Tf: first melting temperature; Th: homogenisation temperature; nd: no data available. * Salinities were calculated using the equation of Bodnar: Salinity (wt. % NaCl equiv.) $= 0.00 + 1.78 \times \theta - 0.0442 \times \theta^2 + 0.000557 \times \theta^3$, where $\theta$ is the freezing point depression (FPD) temperature in °C [7]. FPD = Freezing Point Depression or last melting temperature.

**Table 2.** Summary of microthermometric data at the Penjom gold deposit, Peninsular Malaysia.

| Location | Tf (°C) | FPD (°C) | Th (°C) | Salinity * wt % NaCl Equiv. |
|---|---|---|---|---|
| Penjom Au deposit (Type III) | nd | −4.9 | 236.5 | nd |
|  | nd | nd | 277.3 | nd |
|  | nd | nd | 330.6 | nd |
|  | nd | nd | 220.8 | nd |
|  | nd | nd | 293.8 | nd |
|  | nd | nd | 260.4 | nd |
|  | nd | nd | 295.8 | nd |
|  | nd | nd | 268.5 | nd |
|  | nd | nd | 345.2 | nd |
|  | nd | nd | 302.1 | nd |
| Penjom Au deposit (Type II) | nd | nd | 345.7 | nd |
|  | nd | nd | 282.3 | nd |
|  | nd | nd | 332.4 | nd |
|  | nd | nd | 316.7 | nd |
| Penjom Au deposit (Type I) | −29 | −5.4 | 256 | 8.41 |
|  | nd | −5.6 | 280 | 8.68 |
|  | −44 | −4 | 259.5 | 6.45 |
|  | −28.9 | −4.3 | 256 | 6.88 |
|  | −49 | −2.6 | 236.5 | 4.30 |
|  | −28.1 | −3.9 | 276 | 6.30 |
|  | −28.7 | −4.2 | 302 | 6.74 |
|  | −29.4 | −4.5 | 289 | 7.17 |
|  | −28.8 | −4.8 | 271.6 | 7.59 |

Note: Tf: first melting temperature; FPD = Freezing Point Depression or last melting temperature; Th: homogenisation temperature; nd: no data available. * Salinities were calculated using the equation of Bodnar [7].

Histograms of homogenisation temperatures and salinities are shown in Figures 7 and 8.

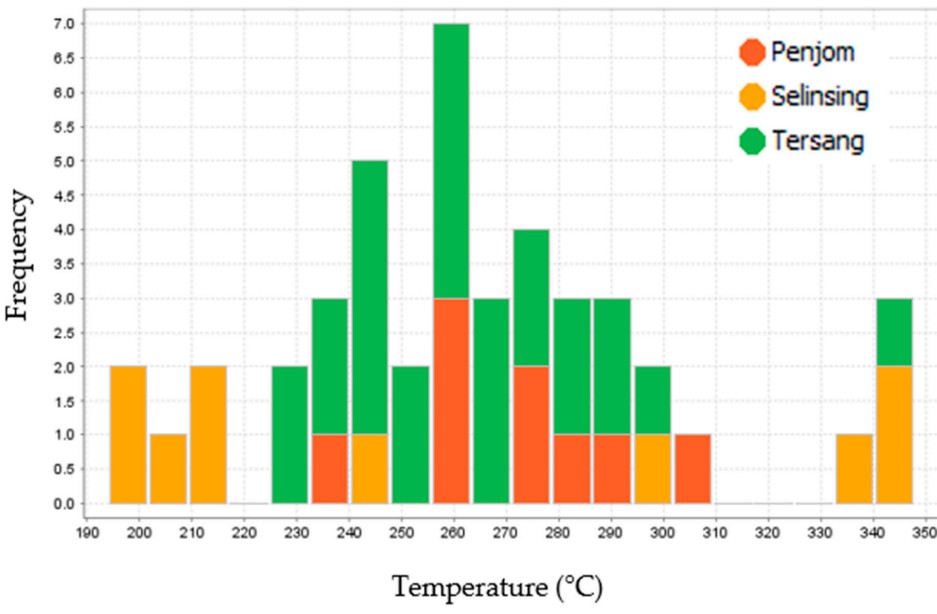

**Figure 7.** Histogram showing homogenisation temperatures of Type I fluid inclusions from the Penjom Selinsing, and Tersang orogenic gold deposits, Peninsular Malaysia.

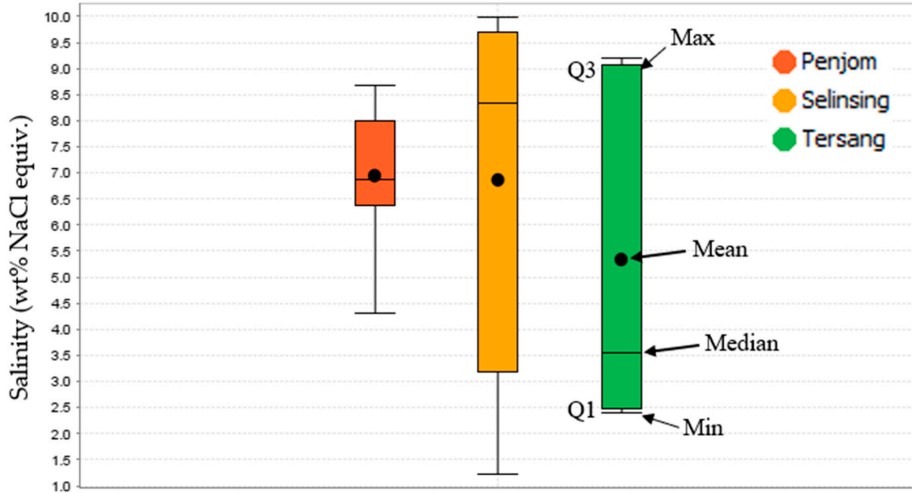

**Figure 8.** Box plots showing salinity values of Type I fluid inclusions. The Penjom deposit has the shortest salinity range. However, Selinsing deposit has the longest salinity range and Tersang deposit has the middle salinity range. Q1 is bottom of box, and Q3 is top of box. Line is the Median value and the black circle is the Mean value. The central box is the middle 50% of data from Q1 to Q3. Note: Min = Minimum; Max = Maximum.

## 6. Composition of Fluid Inclusions

Laser Raman analysis was carried out to determine the composition of fluid inclusions. At the Tersang gold deposit, most inclusions contain pure or nearly pure $CO_2$ (87–100 mol %), and one inclusion was found to contain $CH_4$ (13 mol %) as shown in Table 3. In addition, at Selinsing, most inclusions are $CO_2$-rich (100 mol %) with an exception of one fluid inclusion indicating 5 mol % $N_2$ and 5 mol % $CH_4$. Moreover, at the Penjom gold deposit, most fluid inclusions are $CO_2$-rich (91–100 mol %), whereas one fluid inclusion is $N_2$-rich (100 mol %) and another contains $N_2$ and $CH_4$ as shown in Table 3. The Ternary plot of the fluid inclusion composition for the three existing gaseous components ($CO_2$, $N_2$, and $CH_4$) is shown in Figure 9. Laser Raman Spectra of fluid inclusions are presented in Figure 10. Results of gas composition of fluid inclusions comprising raman shift numbers are presented in supplementary materials (Table S1).

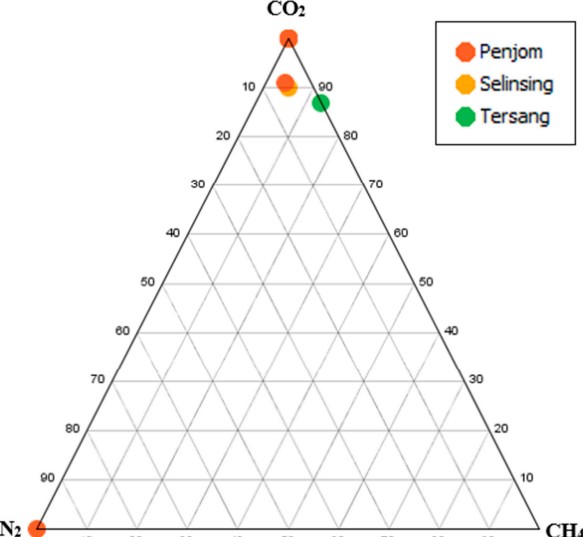

**Figure 9.** Ternary plot of the three gaseous components ($CO_2$, $N_2$, and $CH_4$) showing prevalence of $CO_2$ composition in fluid inclusions from the Penjom, Selinsing, and Tersang orogenic gold deposits, Peninsular Malaysia.

**Table 3.** Laser Raman Spectrometry results for fluid inclusions (Type I) in mineralised quartz veins from the Tersang, Selinsing, and Penjom gold deposits, Peninsular Malaysia.

| Sample | $SO_2$ | $CO_2$ | $N_2$ | $H_2S$ | $C_3H_8$ | $CH_4$ | $C_2H_6$ | $NH_3$ | $H_2$ |
|---|---|---|---|---|---|---|---|---|---|
| Number | mol % | mol % | mol % | mol % | mol % | mol % | mol % | mol % | mol % |
| TER-1 | 0 | 100 | 0 | 0 | 0 | 0 | 0 | 0 | 0 |
| TER-2 | 0 | 100 | 0 | 0 | 0 | 0 | 0 | 0 | 0 |
| TER-3 | 0 | 100 | 0 | 0 | 0 | 0 | 0 | 0 | 0 |
| TER-4 | 0 | 100 | 0 | 0 | 0 | 0 | 0 | 0 | 0 |
| TER-5 | 0 | 100 | 0 | 0 | 0 | 0 | 0 | 0 | 0 |
| TER-6 | 0 | 87 | 0 | 0 | 0 | 13 | 0 | 0 | 0 |
| SEL-1 | 0 | 100 | 0 | 0 | 0 | 0 | 0 | 0 | 0 |
| SEL-2 | 0 | 100 | 0 | 0 | 0 | 0 | 0 | 0 | 0 |
| SEL-3 | 0 | 100 | 0 | 0 | 0 | 0 | 0 | 0 | 0 |
| SEL-4 | 0 | 100 | 0 | 0 | 0 | 0 | 0 | 0 | 0 |
| SEL-5 | 0 | 100 | 0 | 0 | 0 | 0 | 0 | 0 | 0 |
| SEL-6 | 0 | 100 | 0 | 0 | 0 | 0 | 0 | 0 | 0 |
| SEL-7 | 0 | 100 | 0 | 0 | 0 | 0 | 0 | 0 | 0 |
| SEL-8 | 0 | 100 | 0 | 0 | 0 | 0 | 0 | 0 | 0 |
| SEL-9 | 0 | 100 | 0 | 0 | 0 | 0 | 0 | 0 | 0 |
| SEL-10 | 0 | 90 | 5 | 0 | 0 | 5 | 0 | 0 | 0 |
| PEN-1 | 0 | 0 | 100 | 0 | 0 | 0 | 0 | 0 | 0 |
| PEN-2 | 0 | 91 | 5 | 0 | 0 | 4 | 0 | 0 | 0 |
| PEN-3 | 0 | 100 | 0 | 0 | 0 | 0 | 0 | 0 | 0 |
| PEN-4 | 0 | 100 | 0 | 0 | 0 | 0 | 0 | 0 | 0 |
| PEN-5 | 0 | 100 | 0 | 0 | 0 | 0 | 0 | 0 | 0 |
| PEN-6 | 0 | 100 | 0 | 0 | 0 | 0 | 0 | 0 | 0 |
| PEN-7 | 0 | 100 | 0 | 0 | 0 | 0 | 0 | 0 | 0 |
| PEN-8 | 0 | 100 | 0 | 0 | 0 | 0 | 0 | 0 | 0 |
| PEN-9 | 0 | 100 | 0 | 0 | 0 | 0 | 0 | 0 | 0 |

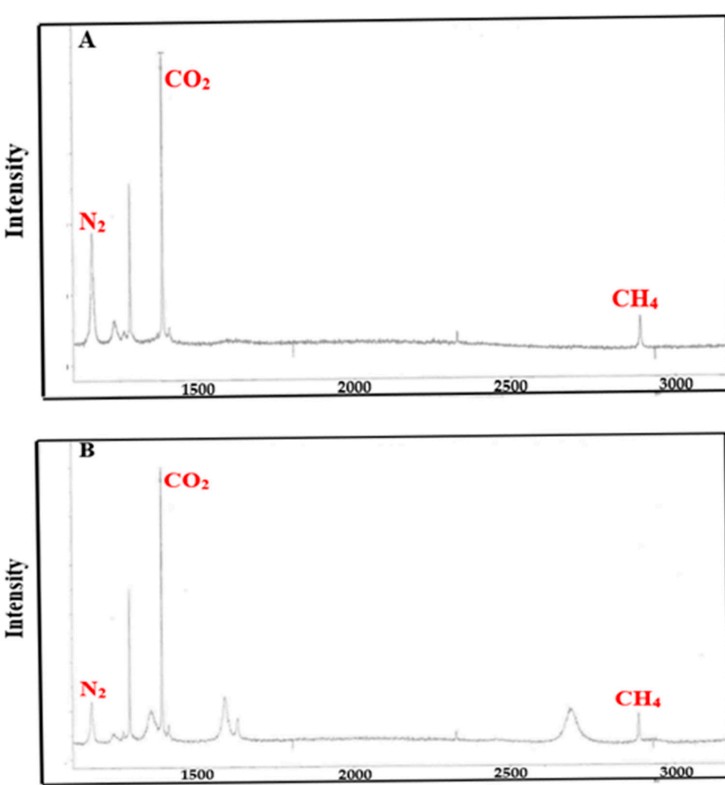

**Figure 10.** *Cont.*

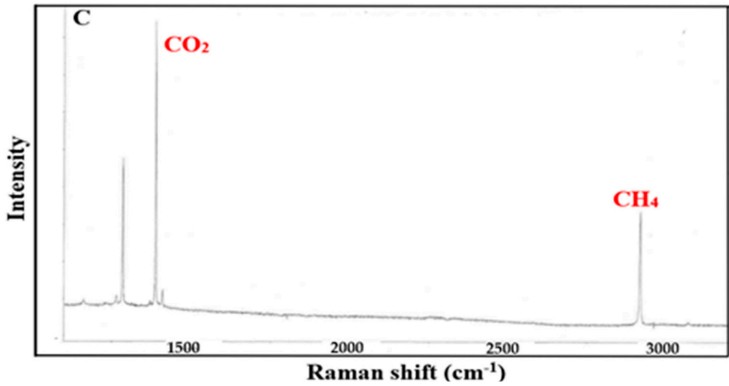

**Figure 10.** Selected laser Raman spectra of fluid inclusion composition. (**A**) Penjom gold deposit. (**B**) Selinsing gold deposit. (**C**) Tersang gold deposit.

## 7. Major and Trace Element Compositions of Igneous Rocks

Igneous rocks that outcrop at the Penjom and Tersang have been analysed and results are shown in Table 4 below. In this table, major elements composition is given in wt %, whereas trace elements composition is in ppm. DL represents the detection limit for each trace element and N/A indicates not applicable for major elements. Samples that have initials PEN come from Penjom gold deposit. Those that have initials TER belong to Tersang gold deposit.

**Table 4.** X-ray fluorescence analysis of the igneous rocks at Penjom and Tersang, Peninsular Malaysia.

| Element | PEN-04 | PEN-14 | PEN-S1 | PEN-S2 | TER-10 | TER-28 | TER-33 | DL |
|---------|--------|--------|--------|--------|--------|--------|--------|-----|
| $SiO_2$ | 81.01 | 74.71 | 56.13 | 76.93 | 78.76 | 74.73 | 77.46 | N/A |
| $TiO_2$ | 0.04 | 0.05 | 0.94 | 0.04 | 0.01 | 0.37 | 0.01 | N/A |
| $Al_2O_3$ | 9.77 | 12.54 | 11.49 | 14.16 | 13.99 | 14.84 | 13.20 | N/A |
| $Fe_2O_3$ | 0.42 | 0.63 | 4.72 | 0.52 | 0.60 | 2.56 | 2.72 | N/A |
| MnO | 0.06 | 0.09 | 0.08 | 0.02 | 0.00 | 0.00 | 0.01 | N/A |
| MgO | 0.15 | 0.25 | 6.57 | 0.26 | 0.23 | 0.30 | 0.18 | N/A |
| CaO | 2.00 | 2.47 | 5.96 | 0.28 | <0.01 | <0.01 | <0.01 | N/A |
| $Na_2O$ | 1.19 | 5.09 | 3.70 | 3.89 | 0.07 | 0.15 | 0.17 | N/A |
| $K_2O$ | 2.25 | 1.14 | 0.06 | 2.15 | 4.13 | 4.12 | 3.69 | N/A |
| $P_2O_5$ | 0.01 | 0.01 | 0.38 | 0.01 | 0.01 | 0.04 | 0.01 | N/A |
| $As_2O_3$ | 0.00 | 0.07 | 0.01 | 0.05 | 0.03 | 0.08 | 0.35 | N/A |
| LOI | 2.86 | 2.99 | 9.51 | 1.60 | 2.08 | 2.56 | 2.54 | N/A |
| Total | 99.76 | 100.04 | 99.54 | 99.90 | 99.91 | 99.75 | 100.33 | N/A |
| Total S | 0.09 | 0.06 | 0.31 | 0.08 | <0.01 | <0.01 | 0.02 | N/A |
| Sc | 6 | 7 | 19 | 5 | 7 | 11 | 9 | 1.5 |
| V | <3 | <3 | 123 | 2 | <3 | 61 | <3 | 3 |
| Cr | 2 | 3 | 567 | 2 | 2 | 17 | <1 | 1 |
| Ni | 2 | 2 | 156 | 1 | 1 | 3 | 4 | 1 |
| Cu | 3 | 3 | 3 | 1 | 2 | 18 | 29 | 1 |
| Zn | 8 | 14 | 53 | 15 | 6 | 21 | 9 | 1 |
| As | 5 | 504 | 44 | 367 | 217 | 565 | 2280 | 3 |
| Rb | 241 | 179 | 6 | 335 | 264 | 212 | 403 | 1 |
| Sr | 85 | 280 | 676 | 194 | 13 | 67 | 8 | 1 |
| Y | 11 | 15 | 26 | 15 | 68 | 24 | 52 | 1 |
| Zr | 33 | 42 | 387 | 44 | 70 | 129 | 55 | 1 |

**Table 4.** *Cont.*

| Element | PEN-04 | PEN-14 | PEN-S1 | PEN-S2 | TER-10 | TER-28 | TER-33 | DL |
|---------|--------|--------|--------|--------|--------|--------|--------|-----|
| Nb | 19 | 27 | 24 | 43 | 13 | 3 | 24 | 0.5 |
| Sn | 10 | 11 | 12 | 17 | 20 | 2 | 35 | 2 |
| Ba | 96 | 51 | 49 | 209 | 220 | 282 | 168 | 4 |
| La | 5 | 5 | 33 | 22 | 18 | 17 | 105 | 2 |
| Ce | 8 | 13 | 81 | *50* | 46 | 36 | 244 | 4 |
| Nd | 4 | 6 | 47 | 26 | 28 | 19 | 131 | 2 |
| Pb | 13 | 123 | 12 | 52 | 64 | 20 | 96 | 1.5 |
| Bi | 4 | 5 | <2 | <2 | 4 | <2 | 37 | 2 |
| Th | 12 | 14 | 55 | 10 | 22 | 10 | 12 | 1.5 |
| U | 26 | 32 | 15 | 8 | 4 | 8 | 8 | 1.5 |

Note: LOI (Loss of ignition), major elements ($SiO_2$ to $As_2O_3$) in wt %, and trace elements (Sc to U) in ppm.

## 8. Discussion

Temperature and salinity determination have shown that the Tersang fluid inclusions have homogenisation temperatures ranging from 232 to 348 °C and low salinities between 2.41 and 9.21 wt % NaCl equiv.). In addition, the Tersang fluid inclusions mostly contain $CO_2$ (98–100 mol %) and traces of $CH_4$. The presence of $CO_2$-rich inclusions suggests that these fluids were probably trapped during metamorphism, which means sourced from metamorphic fluids. The evidence indicates involvement of metamorphic fluids during formation of the Tersang gold deposit.

In comparison, at Selinsing, homogenisation temperatures are between 200 and 340 °C and salinities range from 2.41 to 9.86 wt. % NaCl equiv. which is of similar range to the Tersang fluid inclusions. Fluid inclusions have considerable amounts of $CO_2$ (95–100 mol %), with a minor proportion of $N_2$ (5 mol %). These results are comparable to a previous study on fluid inclusions in quartz-carbonate veins documented in [8] who reported the existence of high temperature (270–280 °C), low salinity and $CO_2$-rich fluids at the Selinsing gold deposit. The authors also suggested that these results showed similarity with other orogenic gold deposits where fluids were introduced into the shear zone carrying gold and other trace elements.

At Penjom, homogenisation temperatures range from 230 to 300 °C and salinities between 4.30 and 8.68 wt % NaCl equiv. Most fluid inclusions are typically $CO_2$-rich (100 mol %) with an exception of one inclusion that is $N_2$-rich (100 mol %). [9] argued that the typical conditions of mineralisation formation in orogenic gold deposits include temperatures of 300–350 °C at pressures of 1–2.5 Kb. This thermal signature is the same in the Tersang, Selinsing, and Penjom gold deposits, which are found in this thermal range as evidenced in this study.

Some authors indicated that the deposits characterized by homogenisation temperatures ranging from 200 to 348 °C, low salinity (<10 wt % NaCl equiv.), and $H_2O$–$CO_2$ ± $CH_4$ ± $N_2$ fluids with depth less than 10 km have similarities to orogenic gold deposits [10,11]. The plot of salinity against homogenisation temperature data for the Type I fluid inclusions from Selinsing and Tersang, as well as the Type I from Penjom is shown in Figure 11A. The figure is adapted from a model (Figure 11B) trying to interpret the fluid evolution [12]. From this model, it is understood that the isothermal mixing of fluid exists if there is contrasting salinity producing the trends 1 and 2 as shown in Figure 11B1–B2. In this trend, there is contrasting salinity trend with no significant variation in homogenisation temperature.

Additionally, surface fluid dilution is the process characterised by the mixing of cooler fluids with less saline fluids as indicated by the trend 3 in Figure 11B3. Cooling is the fluid process, in which there is significant decrease in temperature while salinity almost remains unchanged (Figure 11B4). Boiling involves separation of liquid and vapour is shown by the trend 5. During boiling, there is a decrease of temperatures with increase of salinity (Figure 11B5). The current data for the Selinsing gold deposit indicates that there is decline of temperature and increase in salinity likely suggesting

boiling. At Tersang, temperature does not significantly change but there is increase in salinity probably indicating isothermal mixing.

At a district scale, the plot of homogenisation temperature against salinity (Figure 11) for the Selinsing, Tersang, and Penjom gold deposits likely indicates an isothermal mixing of fluids. This study has shown that hydrothermal fluids involved in the transport and deposition of ore minerals are $CO_2$–$H_2O$ bearing fluids with elevated $CO_2$ content. These fluids are interpreted to have been sourced from metamorphic fluids. The source of the fluids in these deposits is like that of the Sizhuang, orogenic gold deposit, Jiaodong Peninsular in China [13] in which homogenisation temperatures range from 160 to 360 °C and salinity varies between 3.00 and 11.83 wt % NaCl equiv. The interpretation of metamorphic source is to be taken with caution as elevated $CO_2$ contents are not exclusively diagnostic of metamorphic origin of ore-forming fluids.

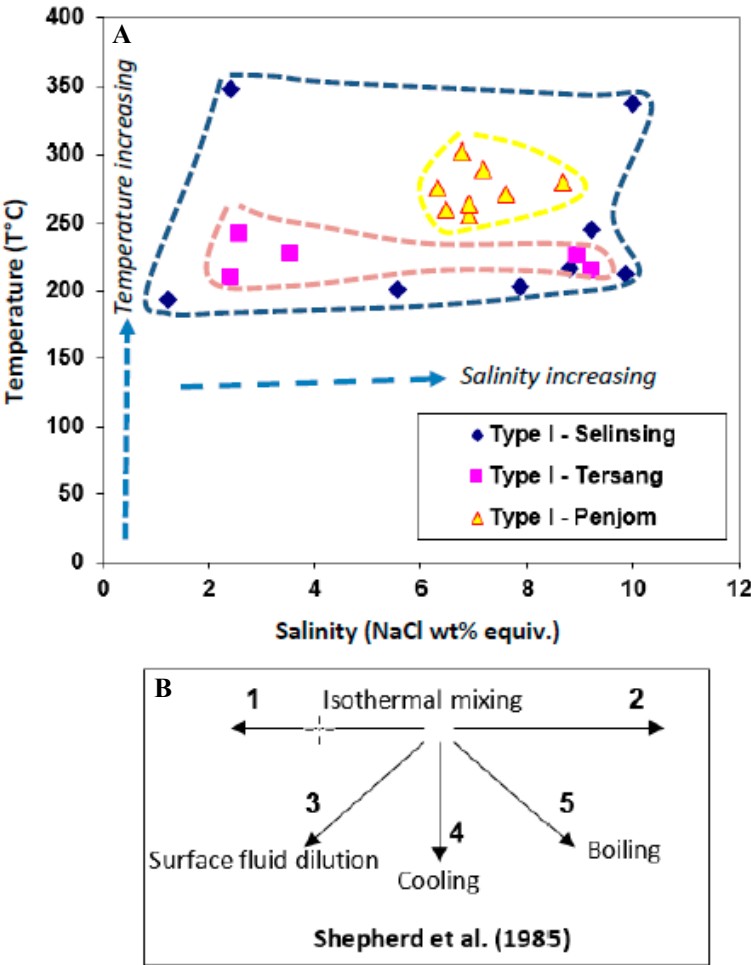

**Figure 11.** Plot of homogenisation temperatures and salinities for the Tersang, Selinsing, and Penjom deposit, Peninsular Malaysia. (**A**) Plot of salinity against homogenisation temperature for inclusion Type I from the Selinsing, Tersang, and Type III from the Penjom gold deposit. (**B**) Model of fluid processing trends. The data in this figure came from the first author Master's thesis at the University of Tasmania.

In terms of salinity, it appears to be two types of fluids, low (2.41–3.55 NaCl wt. % equiv) and saline (8.95–9.25 NaCl wt % equiv.) fluids for the Tersang gold deposit. Variation of salinity data at the Selinsing Au deposit suggests that more than one fluid probably was involved in the formation of Au mineralisation. From field observations, there is a spatial association of igneous rocks and sedimentary/metasedimentary rocks at the Penjom and Tersang deposits (Figures 2 and 3). The igneous

rocks appear in the form of sill and dyke at both the Tersang and Penjom deposits. The igneous rocks have been analysed using XRF and presented on a plot Zr/Ti versus Nb/Y (Figure 12). In this study, the classification of Winchester and Floyd [14] modified by Pearce [15] places the analysed samples in the rhyolite, rhyolite-dacite, and trachyte-andesite compositional fields (Figure 12). The spatial association of igneous rocks at the Tersang and Penjom deposits strongly suggests that a magmatic contribution is possible to the ore-forming fluids. Previous researchers have also documented that igneous intrusions can also produce $CO_2$-rich fluids [16,17].

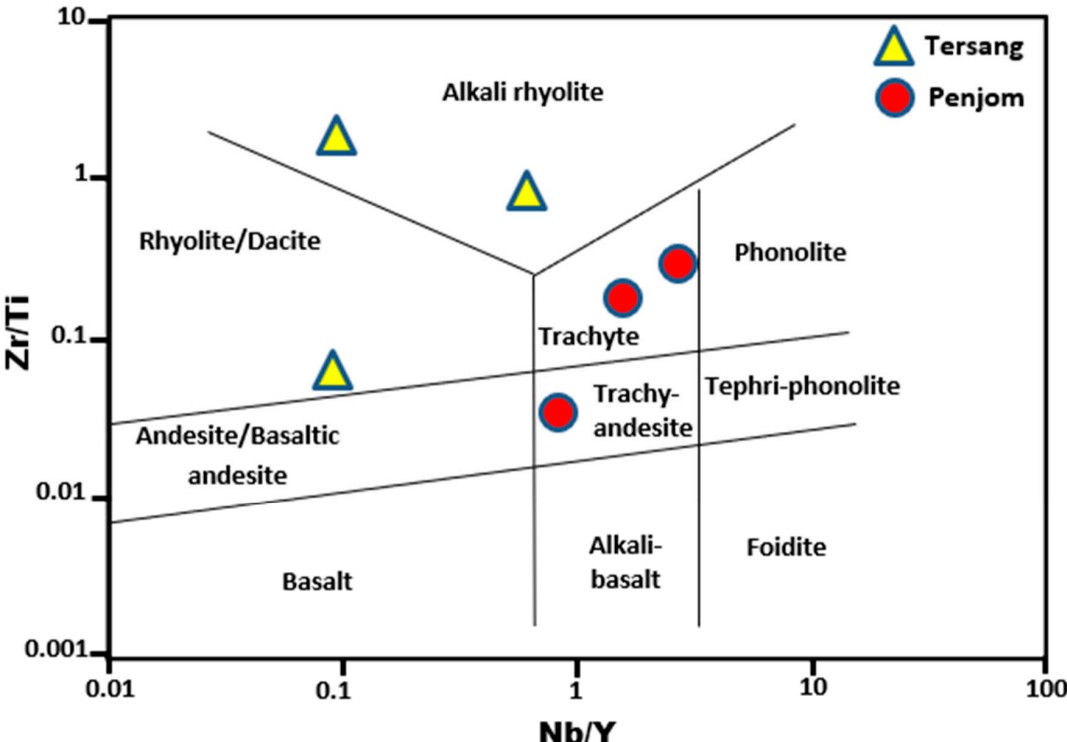

**Figure 12.** Discrimination plot Zr/Ti versus Nb/Y of igneous rocks cropping out at the Tersang and Penjom gold deposits, Peninsular Malaysia.

The evidence seems to suggest a possible contribution of magmatic fluids to the ore system during emplacement of these igneous rocks at the Tersang and Penjom gold deposits. The remarkable decrease in temperature of homogenization and increase in salinity would had favored metal deposition at the Selinsing gold deposit during boiling. Boiling is characterized by the separation of non-volatile solutes into the liquid phase over steam loss [18]. Reference [19] documented that boiling is a likely mechanism that facilitates the deposition of metals from solution at temperatures close to 300 °C. Therefore, the Selinsing area provides promising evidence of finding more undiscovered gold mineralization due to boiling signature with temperatures ranging up to 347.5 °C.

Geologically, most sedimentary rocks at the Penjom and Tersang deposits are associated with magmatic intrusions and contain sheeted veins (Figures 2 and 3) suggesting that these deposits show characteristics of intrusion-related gold deposits. Furthermore, total sulphur calculated (Table 4) is less than 1 wt % also typical of intrusion-related gold deposits, which commonly host low sulphur [20]. In terms of fluid composition, the Penjom and Tersang gold deposits display close affinity to intrusion-hosted sheeted vein systems, which host $CO_2$-rich fluids with minor contents of $CH_4$ and $N_2$. The co-existence of two types of fluids with contrasting salinity at the Tersang gold deposit implies similarity to intrusion-related sheeted quartz veins, such as the Mike lake, intrusion-related deposit in the Yukon Territory, Canada [21].

The Penjom, Tersang, and Selinsing deposits may have formed at deeper environments as they all contain low-salinity (<10 wt % NaCl equiv.), and carbon dioxide-rich aqueous fluids, commonly

immiscible. Previous works have revealed that the $CO_2$-rich fluids may have exsolved from felsic magma hosted in the Phanerozoic, intrusion-related gold deposits [22,23]. This study has also shown the presence of rhyolite formed from felsic magma at the Tersang gold deposit. The Penjom, Tersang, and Selinsing gold deposits may have formed at deeper environments (~>5 km) and are typical of mesothermal gold systems [24]. In previous works, saline inclusion fluids were trapped in quartz veins, which were formed during low-grade metamorphism [25].

Other works at these deposits also indicated proof of low-grade metamorphism supported by exposures of carbonaceous slates or metamorphosed shales, and tuffaceous metasediments [26]. Overall, the evidence indicates involvement of both metamorphic and magma-derived fluids during ore formation. Exploration geologists are encouraged to do further petrographic observations of fluid inclusions across the district to pinpoint favourable zones, which yield $CO_2$-rich fluids. Additionally, future research is needed to investigate fluid inclusions from barren quartz veins in order to compare them with those that are gold-bearing for the purpose of vectoring.

## 9. Conclusions

Microthermometric analysis indicated homogenisation temperatures ranging from 210 to 348 °C (Tersang); between 194 and 348 °C (Selinsing) and from 221 to 346 °C (Penjom). Salinities range from 2.41 to 8.95 wt. % NaCl equiv (Tersang), between 1.23 and 9.98 wt % NaCl equiv (Selinsing), and from 4.34 to 9.34 wt % NaCl equiv (Penjom). Laser Raman studies indicated that at the Tersang gold deposit, most inclusions are either pure or nearly pure $CO_2$-rich (87–100 mol %), except for one inclusion, which contains $CH_4$ (13 mol %). In addition, at the Selinsing gold deposit, most inclusions are $CO_2$-rich (100 mol %). However, an inclusion was found containing $CO_2$ (90 mol %), $N_2$ (5 mol %), and $CH_4$ (5 mol %). Additionally, at the Penjom gold deposit, most fluid inclusions are $CO_2$-rich (91–100 mol %). Additionally, one fluid inclusion is $N_2$-rich (100 mol %) and another one contains $CO_2$ with minor $N_2$ and $CH_4$. The study also indicates a likely mixing of metamorphic, and magmatic fluids with salinity up to 10 wt. % NaCl equiv, the prevalence of $CO_2$-rich inclusions, and the association of sandstone and carbonaceous black shales with magmatic rocks such as rhyolite, rhyolite-dacite, and trachyte-andesite at the Tersang and Penjom deposits in the district.

**Supplementary Materials:** The following are available online at http://www.mdpi.com/2075-163X/10/2/111/s1, Table S1: Gas composition of fluid inclusions by laser raman technique.

**Author Contributions:** Conceptualization, C.M. and K.Z.; Methodology, C.M. and Z.E.; Software, C.M.; Validation, C.M., Z.E., and K.Z.; Formal Analysis, C.M.; Investigation, C.M. and Z.E.; Resources, C.M.; Data Curation, C.M.; Writing-Original Draft Preparation, C.M. and Z.E.; Writing–Review & Editing, C.M. and Z.E.; Visualization, C.M.; Supervision, K.Z.; Project Administration, K.Z. All authors have read and agreed to the published version of the manuscript

**Funding:** This research received no external funding.

**Acknowledgments:** Many thanks to the Staff of Penjom Gold Mine (Malaysia), Monument Mining Limited for allowing access to core samples during fieldwork. Authors are indebted to the team of the project "Ore Deposits of Southeast Asia Project" led by Khin Zaw, for logistic and research support at the University of Tasmania. Our thanks also go to Alexander Cuisson for the preparation of fluid inclusion samples at the Lapidary laboratory, University of Tasmania. Authors are grateful to the University of Tasmania for giving authorization to use the fluid/Melt Inclusion Laboratory at the Sandy Bay Campus for microthermometric studies. Additionally, our gratitude goes to Terry Mernagh for spending enormous amount of time determining the fluid inclusion composition using Laser Raman technique at Geoscience Australia. Invaluable contribution has been made through the peer-review process that greatly improved the content of this article.

**Conflicts of Interest:** The authors declare no conflict of interest.

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
