# Peer review of "Fluid Inclusion Study of the Penjom, Tersang, and Selinsing Orogenic Gold Deposits, Peninsular Malaysia"

_minerals, doi:10.3390/min10020111_

Round 1

Reviewer 1 Report

The most of comments are shown in the manuscript. Here only some general remarks are presented.

Chapter 2 is too small. It is necessary to extend the gold deposits descriptions. Why do you not present the ore mineralogy? Homogenization temperatures don’t reflect the FI trapping temperature because the heterogenization of mineral-forming fluids signs are did not observed. Estimated temperature and pressure values were must be proved by other methods such as mineral or isotopic thermobarometry To evidence the fluid origin, it is necessary to use the stable isotopes study together with the fluid composition data.

Concluding comment: authors are need to provide additional evidences of their conclusions.

Author Response

Reviewer 1

Line 2: Fluid inclusion study of…

Answer: the article title is change to “Fluid inclusion study of the Penjom, Tersang, and Selinsing orogenic gold deposits, Peninsular Malaysia”

Line 54: Description of ore composition

Answer: the description is added to the manuscript.

Line 67: Veinlets

Answer: agreed – the word veins is changed to veinlets

Line 80: Fluid inclusion positions in studied quartz veins (grains).

Answer: positions of fluid inclusions in quartz veins (grains) are shown in the manuscript.

Lines 84-85: deleted sentence

Answer: the sentence was deleted

Line 115: How did you determine the gas concentrations using Raman-spectra? Describe your method please.

Answer: Laser Raman spectroscopy (LRS) was used to quantitatively determine the gaseous components in fluid inclusions. The Raman spectra were typically obtained after 10 accumulations with a 5-s integration time and an estimate 5-cm-1 spectral bandpass. The Raman spectra were calibrated using Ar+ plasma and neon emission lines. Raman detection limits are dependent on instrumental sensitivity and the partial pressure of each gas but are estimated to be about 0.15 MPa for CO2 under the conditions of this study.

Line 145: Figure 3

Answer: captions fixed

Line 151: Figure 4

Answer: captions fixed

Line 253 and Line 256:

How did you calculate the molar percentage of water in fluid inclusions? Describe the procedure.

You must verify these estimates using other methods, such as mineral and isotopic thermometry or paragenetic analysis of the mineral associations.

In this case, the estimates may be unreliable.

Answer: agreed with the reviewer

The section on estimation of depth and pressure has been removed from the revised manuscript. At this stage, we want to publish the temperature and salinity data of fluid inclusions. We might consider doing further work in the future to estimate depth and pressure using the other method proposed by the reviewer.

Line 293: Where are the additional evidences that these fluids were really metamorphic or magmatic?

Answer: other clues of metamorphic and magmatic origin of the fluids are added to the manuscript

Line 295: Absence of fluid boiling is evidence of the differences between fluid inclusions homogenization and trapping temperatures.

Answer: This interpretation was simply deleted as it is related to the estimation of depth of entrapment.

Line 326: To confirm this conclusion it would be better to study the fluid inclusions from the barren veins.

Answer: Agreed – the sentence was revised.

Reviewer 2 Report

The article is about the characterization of the fluid inclusions of samples from three orogenic gold deposits via microthermometric analysis and Raman spectroscopy. From the microthermometric measurements, the freezing point depression (DPF) and the homogenization temperatures (minimum trapping temperatures of the fluid inclusions) (Th) are obtained. From these data, the salinity of the inclusions is determined through the equation used by Bodnar (1983). The results are shown in histograms to display the frequency of each value. Finally, the composition of the gas part of same inclusions is examined with Raman technique.

All the data is used then to estimate, firstly, the entrapment depth by using the software FLINCOR (Brown, 1989), and, secondly, the previous evolution that the fluids may have experienced with the model developed by Shepherd et al. (1985) which extrapolates the potential thermal paths from the relationship between Th versus salinity.

The article, as it is now, it is not recommended for publication for several important reasons:

 1) Certain Figures and Tables are not correctly referred in the text:

  - Figures 2 and 3 are mentioned just at the end, after the rest of the figures.

  - There are two figures 3.

  - Figure 7 is not referred in the text.

 2) Incorrect or insufficient treatment of the data:

  - Data of tables do not correspond with the data plotted at the histograms. In some of them the number of samples differ and/or certain Th’s do not appear in the table.

  - It would be convenient to provide more details about the calculus of the salinity, since it is an important point to can perform a correct discussion, and this calculus is not trivial. The equation used in this article does not correspond to the one used in the article of Bodnar (1983).

  - Raman spectra must be shown.

  - Number of data points in Figure 8 does not match with the number of data shown in the tables.

  - It must be indicated where the elemental data used in figure 10 was obtained. After a research, I have found in the Master thesis of the corresponding author that these data came from an X-Ray fluorescence analysis. This must be explained in the article.

I attach a revised pdf version of the article indicating the points mentioned as comments. The article must be re-written carefully, and experimental information must be added to can be considered for publication.

Author Response

Reviewer 2

Lines 66; 73; 100-102; 111: Type your comments

Answer: it is done in the manuscript.

Lines 180-181: This sentence should be at the end of the next paragraph, after explaining the three deposits, since the data of the histograms correspond to the three sites.

More explanation is needed to know how to read the data of the histograms, since apparently, they do not correspond to the data of the Tables 1 and 2. Please check that the data of the all histograms are correct.

Answer: The sentence was cut and pasted at the end of the paragraph. The homogenisation temperature and salinity data were re-plotted. For a better visualisation of the data, Figure 6 (in previous manuscript) was changed to Figure 7. In addition, Figure 7 (in previous manuscript) was changed to Figure 8. Figure 7 is the histogram plot of all the Th data. Figure 8 is now a box plot of all the salinity data. All is shown in the manuscript. Additionally, Figure 9 is a ternary plot of CO2, N2, and CH4 gases and added to the manuscript. This figure directly shows to the readers the gaseous composition of the fluids without even looking at Table 3.

Lines 194; 207; 211; 213; 216; 267; 312; 315; and 319: type your comments.

Answer: all fixed.

Additional comments:

We are really sorry, the Raman Spectra cannot be found in our folders. They appear to be lost.

Reviewer: The equation used in this article does not correspond to the one used in the article of Bodnar (1983).

Answer: agreed with the reviewer

The reference should be Bodnar (1993) and not Bodnar (1983). It has been corrected.

Round 2

Reviewer 1 Report

After correction the article became more informative and may be published in present form.

Author Response

Thank you very much for your positive comments.

Reviewer 2 Report

The article has been improved, since the data are treated and shown with more accuracy.

Nevertheless, as it is written right now, it cannot be considered for publication if the authors do not have the Raman spectra which demonstrate the composition of the fluid inclusions. The complete part referred to the Raman measurements must be removed because the authors do not have the evidences of the data that they are discussing. However, the article would be acceptable if the authors perform the discussion using references of previous works that have measured the compositions of fluid inclusions of those deposits or others located in close zones and related to them. Due to the lack of the knowledge of the exact composition, the discussion must be wider and open to other possibilities in the case that the fluids can be less rich in CO2.

Same comment for the data shown in Figure 11, the plot Zr/Ti versus Nb/Y. The authors are not giving any information about how these data have been obtained.

Summarizing, the experimental measurements and their analyses must be shown (at least as supplementary information) to can confirm the claims made by the authors or, use previous works that have calculated or measured the data needed to elaborate a good discussion. If this requirement is not accomplished the article cannot be published as it is now.

Author Response

Comments to the version 2 of the manuscript “Characteristics of ore-forming fluids for the Penjom, Tersang, and Selinsing orogenic gold deposits, Peninsular Malaysia” Authors: CHARLES MAKOUNDI *, KHIN ZAW, ZAKARIA ENDUT

Comment 1: The article has been improved, since the data are treated and shown with more accuracy.

Nevertheless, as it is written right now, it cannot be considered for publication if the authors do not have the Raman spectra which demonstrate the composition of the fluid inclusions. The complete part referred to the Raman measurements must be removed because the authors do not have the evidences of the data that they are discussing. However, the article would be acceptable if the authors perform the discussion using references of previous works that have measured the compositions of fluid inclusions of those deposits or others located in close zones and related to them. Due to the lack of the knowledge of the exact composition, the discussion must be wider and open to other possibilities in the case that the fluids can be less rich in CO2.

Answer: We have included laser Raman spectra as shown in Figure 10 (lines 273-298). Peaks of CO2, CH4 and N2 are presented in Figure 10.

Comment 2: Same comment for the data shown in Figure 11, the plot Zr/Ti versus Nb/Y. The authors are not giving any information about how these data have been obtained.

Answer: We have added the X-ray fluorescence analytical method (lines 126-148) and results (lines 301-308) which were used to plot Figure 11.

Summarizing, the experimental measurements and their analyses must be shown (at least as supplementary information) to confirm the claims made by the authors or, use previous works that have calculated or measured the data needed to elaborate a good discussion. If this requirement is not accomplished the article cannot be published as it is now.

Answer: The original spreadsheet that contains the Laser Raman analyses was sent to the editor as supplementary information.
